# CRISPR/Cas14a combined with RPA for visual detection of Marek's disease virus

Zhi-Jian Zhu,[1,2,3] Meng-Li Cui,[2] Yu Liu,[2] Xi-Qiao Yao,[2] Meng-Jie Lu,[4] Ming-Cheng Wang,[2] Jun-He Liu,[2] Jin-Feng Li,[1] En-Zhong Li[2]

**ABSTRACT** Marek's disease, a highly contagious avian immunosuppressive disorder caused by the α-herpesvirus MDV-1, poses a significant threat to poultry health. The development of rapid visual detection methods capable of distinguishing epidemic MDV-1 strains from vaccine strains is crucial for early disease warning, vaccine efficacy evaluation, and precise disease control. We developed a novel isothermal detection system that integrates recombinase polymerase amplification (RPA) with CRISPR/Cas14a technology for the visual identification of epidemic MDV-1 strains. This method operates at a constant temperature of 37°C and allows for either real-time analysis or endpoint visual readout without the need for complex instrumentation. Our results showed no cross-reactivity with Newcastle disease virus, infectious bursal disease virus, MDV-1 vaccine strains, or herpesvirus of turkeys. Plasmid DNA standards were used to determine the sensitivity of the assay, and the detection limit was 24.6 copies/µL. Clinical evaluation using 24 field samples confirmed that the method successfully identified all Marek's disease virus-positive cases, demonstrating its diagnostic reliability. In conclusion, we have developed a rapid, highly specific nucleic acid detection platform for MDV-1 that enables visual readout without complex instrumentation by combining the sensitivity of RPA with the specificity of CRISPR/Cas14a technology, offering promising potential for field-based diagnostics and disease surveillance.

**IMPORTANCE** Marek's disease virus (MDV-1) is a highly contagious and economically important avian pathogen. Existing diagnostic methods are unable to reliably distinguish between epidemic and vaccine strains in field settings, which hampers effective surveillance and evaluation of vaccination programs. To address this challenge, we developed a portable isothermal detection assay that combines recombinase polymerase amplification with CRISPR/Cas14a technology. This approach enables highly sensitive (24.6 copies/µL) and specific visual detection of epidemic MDV-1 strains without cross-reactivity with vaccine strains or related viruses. The assay demonstrated 100% agreement with reference methods when evaluated using clinical samples. As a cost-effective method that avoids the need for complex detection instruments, it offers a practical solution for rapid on-site diagnosis, facilitating enhanced outbreak control and improved poultry health management globally.

**KEYWORDS** RPA, CRISPR/Cas14a, visual detection, Marek's disease virus

Marek's disease (MD) is a highly contagious and economically significant avian lymphoproliferative disorder caused by Marek's disease virus (MDV) (1). It is characterized by immunosuppression, systemic organ dysfunction, and the development of lethal neoplastic transformations in infected chickens, resulting in substantial economic losses to the global poultry industry (2, 3). Based on taxonomic classification, MDV comprises three serotypes: *Gallid herpesvirus 2* (MDV-1), *Gallid herpesvirus 3* (GaHV-3; MDV-2), and *Meleagrid herpesvirus 1* (MeHV-1; herpesvirus of turkeys [HVT])

**Peer Reviewer** Grzegorz Woźniakowski, Nicolaus Copernicus University, Torun, Poland

Address correspondence to En-Zhong Li, enzhongli@163.com, or Jin-Feng Li, jinfengli202512@163.com.

The authors declare no conflict of interest.

See the funding table on p. 12.

(4). Importantly, only field strains of MDV-1 possess oncogenic properties, capable of inducing malignant lymphomas in susceptible chicken populations. Currently, control measures for MD primarily rely on vaccination. The principal vaccine strains used for immunoprophylaxis include CVI988/Rispens and mMDV814, both derived from MDV-1, as well as FC-126, which is derived from HVT (5). Among these, CVI988/Rispens is recognized as the gold-standard attenuated vaccine due to its high efficacy against virulent (v MDV), very virulent (vv MDV), and very virulent plus (vv+ MDV) strains, thus being widely adopted in global MD vaccination programs. However, continuous immune pressure exerted by vaccines has contributed to the evolution of MDV virulence, leading to an increased frequency of vaccine breakthroughs (6, 7). MD outbreaks have become more prevalent in recent years, presenting significant challenges to effective disease control. Therefore, rapid and accurate differential diagnosis of suspected MD cases is essential to enable timely intervention by poultry producers and minimize economic losses.

Common diagnostic methods for MD detection include virus isolation and culture, serological assays, and molecular testing. Virus isolation and identification involve complex and time-consuming procedures, with results often influenced by various confounding factors, particularly interference from vaccine strains. Serological diagnosis, on the other hand, is limited in its ability to reliably differentiate between field strains and vaccine-derived viruses in both infected individuals and clinical cases. Molecular techniques based on nucleic acid amplification, such as polymerase chain reaction (PCR), offer high-throughput and rapid detection capabilities, providing significantly shorter turnaround times and greater accuracy compared to traditional methods. These advantages position PCR as a crucial tool in MD diagnostics, effectively addressing the shortcomings of alternative approaches (8, 9). However, PCR-based detection requires costly equipment and highly trained personnel, which restricts its use in resource-limited settings that lack advanced laboratory infrastructure.

To meet the demands of field diagnostics outside laboratory settings, researchers have developed isothermal nucleic acid amplification techniques, such as loop-mediated isothermal amplification and recombinase polymerase amplification (RPA). RPA technology offers several advantages, including rapid amplification, high sensitivity, strong specificity, and the ability to operate at a constant temperature within the range of 37°C–42°C (10). Since its introduction, RPA has been widely applied in the detection of various viruses, such as severe acute respiratory syndrome coronavirus (SARS-CoV) (11), monkeypox virus (MPXV) (12), and African swine fever virus (13). Zeng et al. designed specific primers and probes targeting the *meq* gene of MDV for detection, achieving high specificity and sensitivity (14). However, RPA's relatively high tolerance for primer binding site mismatches may result in non-specific amplification among closely related species, limiting its ability to distinguish between vaccine strains such as CVI988/Rispens. Given this limitation, combining rapid RPA amplification with the precise gene-editing capabilities of CRISPR/Cas systems has emerged as a promising direction in the development of current and future pathogen diagnostic technologies (15–17).

The CRISPR/Cas system, a transformative tool for gene editing, encompasses a diverse array of effector proteins that have unlocked novel strategies for molecular detection (18). Among these, the CRISPR/Cas14 family includes the smallest known RNA-guided nucleases, with the Cas14a effector typically possessing a relative molecular mass between 40 and 70 kDa (19). Guided by a single guide RNA (sgRNA), Cas14a recognizes and cleaves single-stranded DNA (ssDNA) targets without requiring a protospacer adjacent motif (PAM) (19). Upon target binding, it activates its trans-cleavage activity, leading to non-specific degradation of ssDNA reporters. A key feature of Cas14a is its high sensitivity to single-nucleotide mismatches within the target sequence, which is often linked to a proposed "seed region" in the sgRNA, where full complementarity is critical for activation (19, 20). This property makes Cas14a particularly well suited for distinguishing single-nucleotide polymorphisms (SNPs). It should be noted, however, that mismatch tolerance may not be absolute and can vary, depending

on the mismatch position. Recent studies suggest that certain single mismatches may not fully abolish Cas14a activity, and effective discrimination may sometimes require multiple mismatches, indicating a mechanism with nuanced, position-dependent behavior that shows parallels to the RNA-targeting Cas13a system (21, 22). In recent years, the integration of Cas14a's trans-cleavage activity with various signal amplification strategies has enabled rapid, sensitive pathogen detection, underscoring its strong potential for diagnostic applications (23, 24). Building on this general high-fidelity recognition capability, we developed an enhanced Cas14a-based assay that combines RPA, λ exonuclease-mediated strand generation, and CRISPR/Cas14a collateral cleavage activity. This integrated approach allows for sensitive, specific, instrument-free, and visually interpretable detection of MDV-1 by targeting the conserved *meq* gene (Fig. 1).

## MATERIALS AND METHODS

### Viruses and clinical samples

The MDV-1 Md5 strain was obtained from the laboratory stock. The MDV-1 vaccine strain CVI988/Rispens and HVT Fc-126 are commercially available vaccines provided by Merck Animal Health. The Newcastle disease virus (NDV) was supplied by Zhumadian Animal Disease Prevention and Quarantine Center, and the infectious bursal disease virus (IBDV) B87 was purchased from the Guangdong Epidemiology Prevention and Control Center. A total of 24 tissue samples were obtained from diverse poultry farms situated in Henan Province. During the sample collection procedure, the animals were humanely euthanized via intraperitoneal injection of pentobarbital sodium at a dosage of 150 mg/kg.

### DNA/RNA extraction

The nucleic acid of viruses and clinical samples were extracted using the TIANamp Virus DNA/RNA kit (no DP315; Tiangen Biotech, Beijing, China) according to the manufacturer's instruction and stored at −80°C until used.

### Generation of DNA standard

The open reading frame of the *meq* gene from the MDV-1 epidemic strain (international reference strain Md5, GenBank accession no. AF243438.1) and the vaccine strain CVI988/Rispens (GenBank accession no. DQ530348.1) was amplified using the forward primer 5′-TGCTGGAATGTTAAGAATAAATTCCGCAC-3′ and the reverse primer 5′-TTATCTC

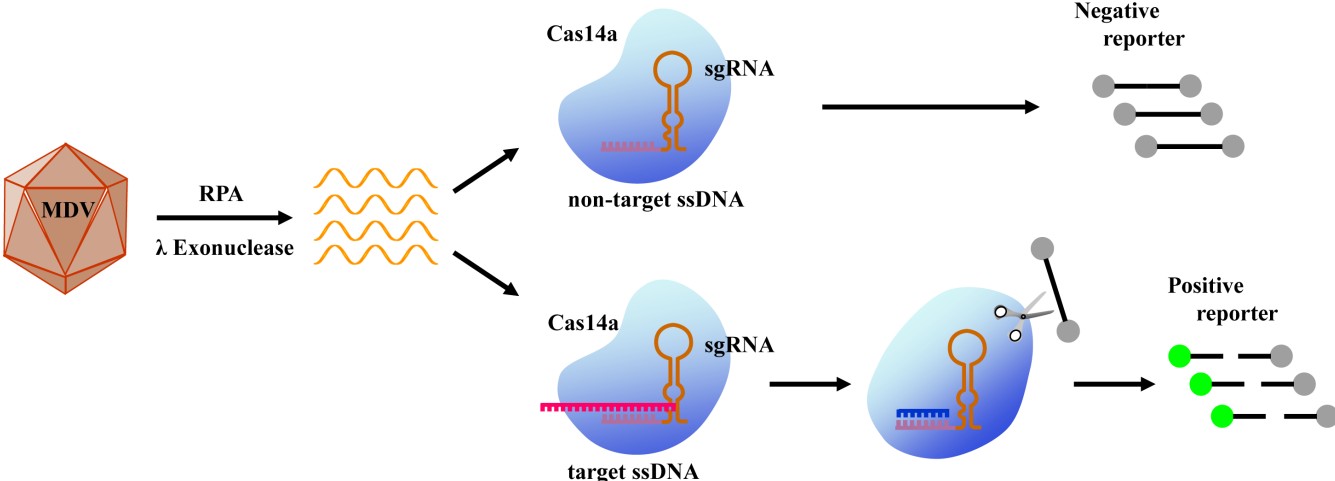

**FIG 1** Schematic illustration of the MDV detection method utilizing the integration of RPA and CRISPR/Cas14a technologies.

ATACTTCGGAACTCCTGG-3′, respectively. The expected amplicon sizes were 1,020 bp for Md5 and 1,197 bp for CVI988/Rispens. The S-meq and L-meq genes amplified from Md5 and CVI988/Rispens were purified using the EasyPure PCR Purification Kit (no. EP101-01; TransGen Biotech, Beijing, China) and subsequently cloned into the pMD-19T vector (Takara, Dalian, China), resulting in the recombinant plasmids pMD-19T-S-meq and pMD-19T-L-meq. Positive clones were selected and sequenced for further processing. Plasmids were extracted using the SanPrep Plasmid Mini-Preparation Kit (no. B518191; Sangon Biotech, Shanghai, China), according to the manufacturer's instructions. DNA concentration was measured by spectrophotometry, and the copy number was calculated prior to being used as a DNA standard for sensitivity analysis.

## RPA primer design and screening

Twenty-one nucleotide sequences of the reference MDV *meq* genes were aligned using DNASTAR software to identify conserved regions (Fig. S1). Based on the highly conserved G→T single-nucleotide polymorphism at position 211 of the *meq* gene between vaccine and epidemic strains, we designed RPA primer pairs specifically targeting this locus using Primer Premier 5.0 software, ultimately obtaining three primer pairs (comprising three forward and three reverse primers). All reverse primers had phosphorylated 5′ ends. The RPA primers were synthesized by Sangon Biotech (Table 1).

The RPA reaction was conducted using a TwistAmp Basic kit (no. TABAS03KIT; TwistDX, Cambridge, UK). A 1 µL aliquot of positive control plasmid pMD19-T-S-*meq* served as the template and was amplified with different primer combinations (Table 1), according to the manufacturer's instructions. The reaction was carried out in a 50 µL system at 37°C for 20 min. The resulting RPA amplicons were separated by electrophoresis on a 1% agarose gel for analysis.

## sgRNA preparation and Cas14a detection

For sgRNA preparation targeting the single-nucleotide polymorphism at position 211 of the *meq* gene, DNA templates of sgRNA were appended with the T7 promoter sequence and synthesized as primers by Sangon Biotech (Table 1). The two primers were annealed into double-stranded DNA using the Annealing Buffer for DNA Oligos (no. D0251, Beyotime). The double-stranded DNA was purified through gel extraction. Using the T7 high-efficiency transcription kit (no. JT101-01, TransGen Biotech), we transcribed the DNA into sgRNA via overnight incubation at 37°C. The resulting sgRNA was then purified using the EasyPure RNA purification kit (no. ER701-01, TransGen Biotech) following the manufacturer's protocol and was stored at −80°C.

TABLE 1   Sequences of RPA primers, sgRNA FQ-polyT, and LF-polyT in this study[a]

| Name | Sequence (5′–3′) |
| --- | --- |
| meq-F1 | AAAGGAAAAGTCACGACATCCCCAACAGCC |
| meq-F2 | GCGCTATGCCCTACAGTCCCGCTGACGATC |
| meq-F3 | GGAGGAGAAACAGAAGCTGGAAAGGAGGAG |
| meq-R1 | p-TTGTCTACATAGTRCGTCTGCTYCCTGCGTC |
| meq-R2 | p-CATAGTRCGTCTGCTYCCTGCGTCTTCTCC |
| meq-R3 | p-ATGTGGAGCGTTAGGTTCATCCGGTGAGGG |
| sgRNA | UUCACUGAUAAAGUGGAGAACCGCUUCACCA |
| | AAAGCUGUCCCUUAGGGGAUUAGAACUUGAG |
| | UGAAGGUGGGCUGCUUGCAUCAGCCUAAUGU |
| | CGAGAAGUGCUUUCUUCGGAAAGUAACCCUC |
| | GAAACAAAUUCAUUUGAAAGAAUGAAGGAAU |
| | GCAACUcuccgagcggcgucacgau |
| FQ-polyT | 6-FAM/TTTTTTTTTTTTT/BHQ1 |
| LF-polyT | 6-FAM/TTTTTTTTTTTTT/Bio |

[a]p, 5′-phosphorylation of primers; R, degenerate base (A or G); Y, degenerate base (T or C).

Cas14a detection was carried out according to the manufacturer's instructions (no. C620038-1000, Sangon Biotech). The 50 µL Cas14a reaction mixture contained 5 µL RPA amplification products, 25 nM sgRNA, 50 nM Cas14a, 2 µL RNase inhibitor (no. M0314S, New England Biolabs), 100 nM ssDNA reporter (Table 1), 0.6 µL λ Exonuclease (no. M0262S, New England Biolabs), and Cas14a detection buffer (1×). Fluorescence kinetics were monitored at 37°C over a 2-h period using an excitation wavelength of 485 nm and an emission wavelength of 520 nm, with fluorescence readings recorded every 4 min.

## Cas14a detection with lateral flow assay

For lateral flow-based Cas14a detection, the ssDNA reporter was replaced with a synthesized FAM-ssDNA-biotin reporter at a final concentration of 100 nM in a 50 µL Cas14a reaction system. The Cas14a reaction was performed at 37°C for 1 h. Subsequently, the CRISPR Cas12/13 HybriDetect Dipstick (no. JY0301, Warbio) was immersed in the solution. Following a 5-min incubation period, images of the dipsticks were captured using a camera. To minimize potential non-specific background signals on the lateral flow strip, it is recommended to use freshly prepared reagents and strictly adhere to the specified incubation times.

## Clinical sample detection

The performance of the RPA-CRISPR/Cas14a assay was assessed using a panel of 24 clinical samples. For comparative analysis, all samples were simultaneously analyzed using a PCR assay targeting the *meq* gene. The PCR reaction was carried out in a 20 µL volume consisting of 10 µL of 2× PCR Mix (Takara), 2 µM each of the forward primer (5′-TGCTGGAATGTTAAGAATAAATTCCGCAC-3′) and reverse primer (5′-TTATCTCATACTTCG GAACTCCTGG-3′), and 25 ng of DNA template. Amplification products were analyzed by electrophoresis on a 1% agarose gel.

## Comparison of RPA-CRISPR/Cas14a with qPCR for clinical sample detection

Since no established qPCR assay can differentiate between epidemic and vaccine strains of MDV-1 by targeting the *meq* gene, we selected the conserved *pp38* gene for absolute viral quantification. Plasmid standards, pMD19-T-OVO (chicken *ovo* gene) and pMD19-T-pp38, were quantified spectrophotometrically, and copy numbers were calculated. Ten-fold serial dilutions of the plasmids, ranging from $10^8$ to $10^1$ copies/µL, were used to generate standard curves. The qPCR reaction mixture (25 µL) contained 0.4 µM of each primer (pp38-FP and pp38-RP), 0.2 µM of the pp38 probe (FAM-labeled, Table 2), and 1× ABsolute Blue qPCR Low Rox Mix (no. AB1318B; Thermo Fisher Scientific, UK). For the duplex reaction, which simultaneously quantified the *pp38* and chicken ovotransferrin (*ovo*) genes, the mixture additionally included 0.4 µM of each *ovo* primer and 0.2 µM of the *ovo* probe (VIC-labeled, Table 2). All reactions were performed in triplicate on an ABI 7500 System (Applied Biosystems) under the following conditions: 95°C for 15 min, followed by 40 cycles of 95°C for 15 s and 60°C for 1 min. Standard curves were constructed by plotting the threshold cycle (Ct) values against the logarithm of the

**TABLE 2** Oligonucleotide primers and probes used in real-time PCR

| Target | Primer/probe | Sequence (5′–3′) | Amplicon size (bp) |
|---|---|---|---|
| pp38 | pp38-FP | GAGCTAACCGGAGAGGGAGA | 99 |
| | pp38-RP | CGCATACCGACTTTCGTCAA | |
| | pp38-probe | CTCCCACTGTGACAGCC | |
| | | (5′-FAM label, 3′-BHQ1 quencher) | |
| OVO | OVO-FP | CACTGCCACTGGGCTCTGT | 71 |
| | OVO-RP | GCAATGGCAATAAACCTCCAA | |
| | OVO-probe | AGTCTGGAGAAGTCTGTGCAGCCTCCA | |
| | | (5′-VIC label, 3′-TAMRA label) | |

template copy number. The efficiency, linearity ($R^2 > 0.99$), and distinct melt curves were assessed to establish the performance of the assay.

Total DNA from the 24 clinical samples was adjusted to a uniform concentration of 25 ng/μL. Absolute quantification of *pp38* and *ovo* copy numbers was performed using the respective standard curves. To account for variations in cellular input, the viral load was normalized to the cellular content and expressed as copies per $10^6$ cells using the following formula: viral copies per $10^6$ cells = $(2 \times [pp38] \times 10^6) / [ovo]$, where [pp38] and [ovo] represent the measured copy numbers per μL for the viral and reference genes, respectively. The factor of 2 accounts for the diploid nature of the chicken genome.

## RESULTS

### Optimal RPA primer set selection

In this study, three forward and three reverse primers were designed and systematically evaluated across all possible combinations, yielding nine distinct primer sets. These were assessed using real-time RPA followed by agarose gel electrophoresis. The results showed that all primer combinations effectively recognized and amplified the target sequence of MDV. However, comparative analysis of the gel profiles revealed that the meq-F3/R3 set produced a faint non-specific band above the expected amplicon, whereas the meq-F1/R3 combination yielded a single, specific product (approximately 395 bp) with no detectable off-target amplification (Fig. 2). The specificity of the amplification product generated by the selected meq-F1/R3 primer set was further confirmed by Sanger sequencing, which verified a perfect match to the target sequence within the meq gene of the MDV-1 Md5 strain (data not shown). Consequently, the meq-F1/R3 primer pair was identified as the optimal RPA primer system and was utilized in subsequent experiments.

### Performance evaluation of the sgRNA-Cas14a fluorescence detection

To evaluate the sensitivity and specificity of the designed sgRNA in recognizing the *meq* gene target sequence among prevalent MDV strains, the complete Cas14a reaction system and the Cas14a reaction system lacking sgRNA were comparatively analyzed. The positive control plasmid pMD-19T-S-meq was used as the template in the RPA assay, and its amplification products were introduced into either the complete Cas14a reaction system or the system without sgRNA for fluorescence kinetic analysis. As shown in Fig. 3, the relative fluorescence units (RFU) in the complete Cas14a system increased rapidly over time until the peak value, whereas no significant fluorescence

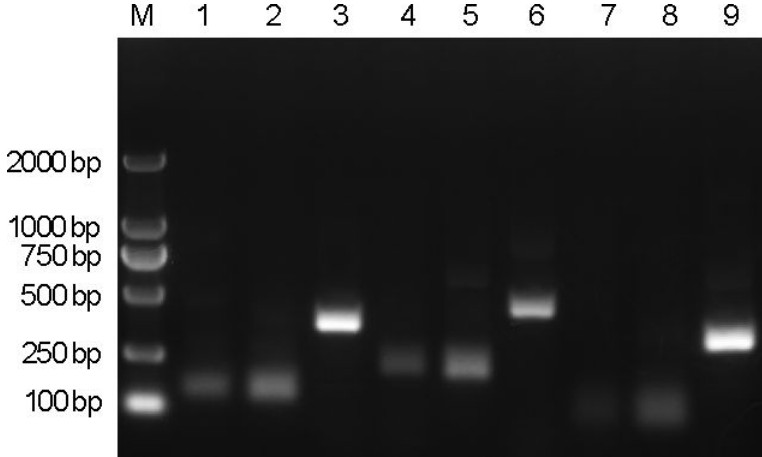

**FIG 2** Agarose gel electrophoresis analysis of RPA-amplified products targeting the *meq* gene of MDV-1 strain. M, DNA marker; 1, meq-F1/meq-R1; 2, meq-F1/meq-R2; 3, meq-F1/meq-R3; 4, meq-F2/meq-R1; 5, meq-F2/meq-R2; 6, meq-F2/meq-R3; 7, meq-F3/meq-R1; 8, meq-F3/meq-R2; 9, meq-F3/meq-R3.

signal was detected in the system without sgRNA. To further verify the specificity of sgRNA-mediated recognition of the *meq* target sequence, RPA products derived from the negative control plasmid pMD-19T-L-meq were subjected to the same fluorescence-based Cas14a detection system. Notably, only the pMD-19T-S-meq plasmid exhibited a time-dependent increase in RFU, while both the negative control and pMD-19T-L-meq groups remained non-reactive (Fig. 3). These results demonstrate that the sgRNA-Cas14a fluorescence detection system enables highly specific identification of MDV-1 epidemic strains.

## Specificity and sensitivity of the enhanced Cas14a fluorescence detection

The specificity of the enhanced Cas14a fluorescence detection was assessed using the MDV international standard virulent strain (Md5), vaccine strains (CVI988/Rispens and HVT), as well as other avian viruses, including NDV and IBDV. Nucleic acid extracts from the different viruses were adjusted to an equimolar concentration of $10^6$ copies/µL prior to being used as templates in the RPA-CRISPR/Cas14a assays. As shown in Fig. 4A, a rapid and strong fluorescent signal was observed exclusively for the Md5 strain. In contrast, no detectable fluorescence was generated in response to the MDV-1 vaccine strains, other tested viruses, or the negative controls. These findings indicate that the proposed detection method possesses high specificity and does not exhibit cross-reactivity with other pathogens, thereby enabling accurate differentiation of MDV-1 epidemic strains.

The analytical sensitivity of the enhanced Cas14a fluorescence detection assay was evaluated using a 10-fold serial dilution of the pMD-19T-S-meq positive standard plasmid. As illustrated in Fig. 4B, the system was capable of detecting target DNA across seven orders of magnitude, ranging from $2.46 \times 10^8$ copies/µL to $2.46 \times 10^1$ copies/µL. These data showed that the limit of detection for the enhanced Cas14a fluorescence detection method is 24.6 copies/µL.

## Performance assessment of the enhanced Cas14a lateral flow detection

To enable rapid on-site detection with visual readout, we developed an enhanced lateral flow detection (LFD) system based on the Cas14a protein. This assay employs a FAM-labeled ssDNA-biotin reporter for Cas14a-mediated target recognition. In negative samples, gold-conjugated anti-FAM antibodies bind to intact reporter molecules, forming complexes that are captured by biotin ligands at the control line (C line). In positive samples, Cas14a-mediated cleavage of the FAM-ssDNA-biotin reporter separates the FAM moiety from the biotin anchor. The liberated FAM fragments then bind to the

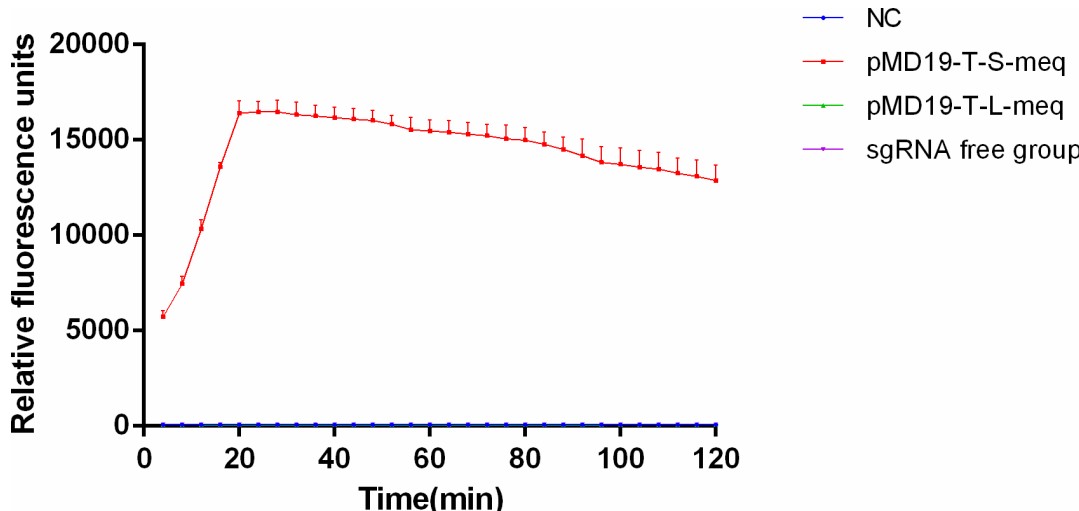

**FIG 3** Analysis of positive standard plasmid pMD-19T-S-meq by RPA-CRISPR/Cas14a fluorescence detection (*n* = 3 technical replicates; values represent mean ± SEM).

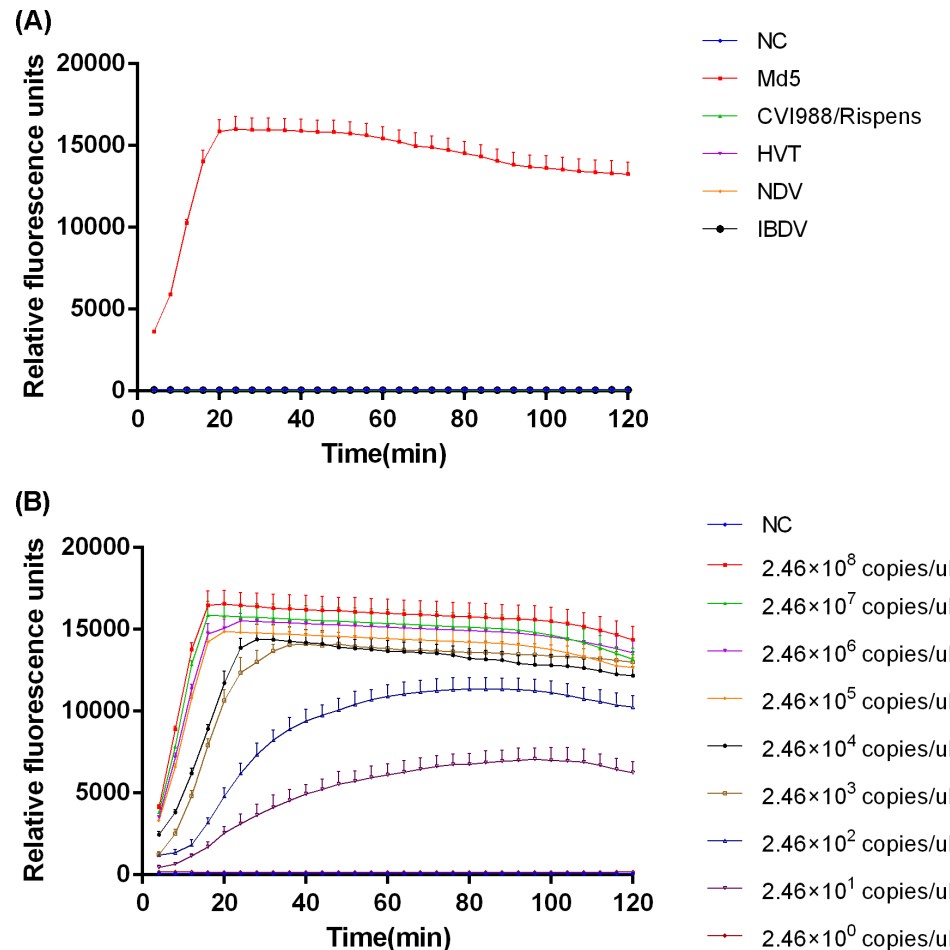

**FIG 4** Specificity and sensitivity of RPA-CRISPR/Cas14a fluorescence detection. (A) Specificity of RPA-CRISPR/Cas14a fluorescence detection. All viral nucleic acid templates were tested at an equimolar concentration of $10^6$ copies/μL ($n$ = 3 technical replicates; values represent mean ± SEM). (B) Sensitivity of RPA-CRISPR/Cas14a fluorescence detection ($n$ = 3 technical replicates; values represent mean ± SEM).

gold-conjugated anti-FAM antibodies, forming complexes that are no longer captured by the biotin ligands at the C line. Instead, these free gold-antibody-FAM complexes are captured by immobilized ligands at the test line, leading to the visible accumulation of colloidal gold and the appearance of a distinct test band, while the signal intensity at the control line decreases accordingly. When applied to detect pMD-19T-S-meq and pMD-19T-L-meq plasmids, the assay specifically identified pMD-19T-S-meq, as indicated by a clearly visible test line (Fig. 5). These results confirm that the enhanced Cas14a-LFD enables visual discrimination of MDV-1 epidemic strains.

## Specificity and sensitivity of the enhanced Cas14a lateral flow detection

To evaluate the specificity of the enhanced Cas14a-based lateral flow assay (LFA), various viruses, including the Md5 strain, vaccine strain CVI988/Rispens, HVT, NDV, and IBDV, were analyzed. The results showed that positive signals were exclusively detected on the lateral flow strips corresponding to the Md5 strain (Fig. 6A). To assess the sensitivity of the enhanced Cas14a lateral flow detection method, 10-fold serial dilutions of the pMD-19T-S-meq plasmid described previously were tested (Fig. 6B). The detection limit was determined to be 24.6 copies/μL, which is consistent with the sensitivity observed in the enhanced Cas14a fluorescence detection method. Therefore, the enhanced Cas14a

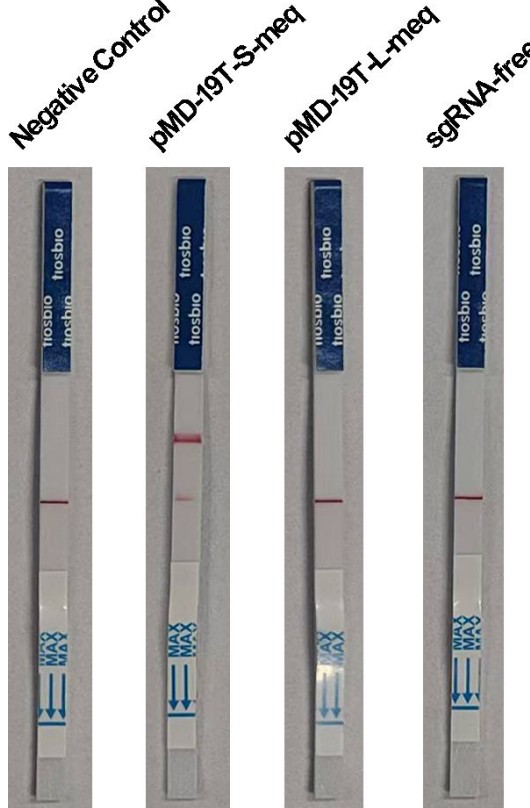

**FIG 5** Analysis of positive standard plasmid pMD-19T-S-meq by RPA-CRISPR/Cas14a lateral flow detection.

lateral flow assay can be effectively utilized for specific and sensitive detection of MDV-1 epidemic strains.

### Enhanced Cas14a detection in clinical samples

To evaluate the clinical performance of the RPA-CRISPR/Cas14a system, 24 field samples were analyzed in parallel using the enhanced Cas14a fluorescence assay, the Cas14a LFA, and conventional *meq*-PCR. Both Cas14a-based methods produced identical results, categorizing 17 samples as MDV-1-positive and 7 as negative (Table 3; Fig. S2 and S3). These results showed 100% concordance with conventional PCR (Table 3; Fig. S4), confirming the diagnostic reliability of our platform for strain-specific detection.

To provide a quantitative assessment of viral load and enable direct comparison with the Cas14a readouts, all samples were subjected to TaqMan qPCR targeting the *pp38* gene. The qPCR assay exhibited excellent performance, with a strong linear standard curve [$R^2 = 0.995$, $Ct = -3.51 \times \log_{10}(Q) + 36.460$] and a detection limit of 10.2 copies/µL. The qPCR results were in complete agreement with the Cas14a assays, with no false positives or negatives (Table 3). The normalized viral loads in positive samples ranged from $5.2 \times 10^3$ to $1.3 \times 10^6$ copies per $10^6$ cells (Table S1), demonstrating the platform's ability to detect MDV-1 across a wide dynamic range.

In terms of detection speed, the fluorescence-based Cas14a assay generated positive signals within 20–40 min, with higher viral loads leading to faster signal onset. For lateral flow detection, the complete process—comprising a 60-min Cas14a reaction followed by a 5-min dipstick incubation—yielded clearly interpretable visual results within a total of approximately 65 min (Fig. S3).

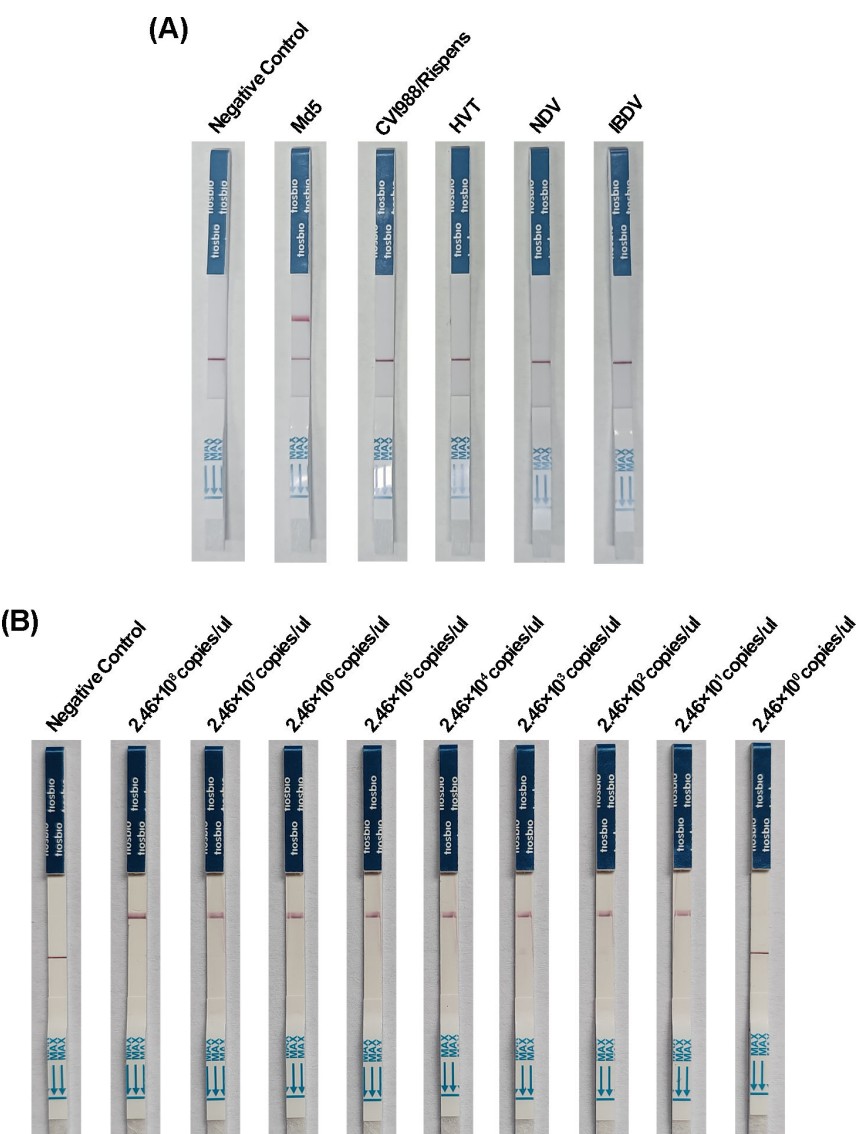

**FIG 6** Specificity and sensitivity of RPA-CRISPR/Cas14a lateral flow detection. (A) Specificity of RPA-CRISPR/Cas14a lateral flow detection. All viral nucleic acid templates were tested at an equimolar concentration of $10^6$ copies/µL. (B) Sensitivity of RPA-CRISPR/Cas14a lateral flow detection.

## DISCUSSION

Nucleic acid-based techniques have become powerful tools for MDV identification due to their high sensitivity. Among these, RPA has gained increasing attention in MDV diagnostics owing to its minimal reliance on sophisticated instruments and stringent laboratory conditions. However, the high tolerance of RPA to base-pair mismatches limits its ability to differentiate between pathogenic MDV strains and vaccine strains such as CVI988 or mMDV814. Given the widespread mandatory vaccination against MDV in poultry farming across China, the direct application of RPA-based methods in clinical diagnosis poses a significant challenge. Therefore, there is an urgent need to develop an MDV detection assay that retains high sensitivity while offering enhanced specificity.

In this study, we established a novel nucleic acid detection system for MDV by integrating RPA with CRISPR/Cas14a technology. The *meq* gene, a unique molecular marker of MDV-1 whose encoded protein plays a critical role in viral oncogenicity and pathogenesis, was selected as the detection target (25, 26). During assay development,

**TABLE 3** Comparison of virus detection among RPA-CRISPR/Cas14a, PCR, and qPCR in clinical samples

| Assay | Number of samples | | |
|---|---|---|---|
| | Positive | Negative | CVI988/814 |
| RPA-CRISPR/Cas14a fluorescence detection | 17 | 7 | /[a] |
| RPA-CRISPR/Cas14a lateral flow detection | 17 | 7 | / |
| PCR | 17 | 7 | 10 |
| qPCR | 17 | 7 | / |

[a]"/" indicates a negative detection result when the sample containing the vaccine strain CVI988/814 was tested using the corresponding method.

we performed multiple sequence alignment of the *meq* gene from 21 MDV reference strains available in the NCBI database to identify highly conserved regions suitable for RPA primer design. Notably, within one such conserved region, we identified a G→T SNP that distinguishes vaccine strains from epidemic strains. This SNP served as the basis for designing a strain-specific sgRNA.

The high specificity of our assay, which enables single-base discrimination between epidemic and vaccine strains, is fundamentally attributed to the precise target recognition by the Cas14a-sgRNA complex. Our successful discrimination of the G→T SNP at position 211 of the *meq* gene aligns with the documented sensitivity of Cas14a to mismatches, particularly within the presumed seed region of the guide RNA. Nevertheless, we recognize that the effect of mismatch location on Cas14a activity is complex and position dependent, as some studies have reported that certain single mismatches do not always lead to complete loss of cleavage activity (21, 22). In our system, the strategic design of the sgRNA targeting this critical SNP, combined with the inherent selectivity of the upstream RPA amplification, established a dual-check mechanism that ensured robust strain-specific detection. Future optimization efforts could include systematic profiling of mismatch tolerance patterns for specific diagnostic targets to further refine sgRNA design. Performance evaluation demonstrated that the RPA-CRISPR/Cas14a system exhibited excellent detection characteristics for MDV-1 epidemic strains, with a sensitivity of 24.6 copies/μL. Compared with the RPA method reported by Zeng et al. (14), which has a detection limit of 100 copies per reaction, our system achieved comparable sensitivity while maintaining the high sequence-specific recognition capability inherent to CRISPR-based technology.

The RPA-CRISPR/Cas14a detection system also demonstrated strong performance in clinical evaluation. Among 24 clinical specimens collected from poultry farms across Henan Province, China, 17 tissue samples tested positive for MDV-1, showing complete concordance with conventional PCR results, thereby confirming the reliability of our system for the clinical diagnosis of Marek's disease. Notably, the Cas14a protein not only supports fluorescence-based detection but also can be adapted for visual readout using lateral flow strips, offering the dual advantages of laboratory-compatible quantification and field-deployable rapid screening. Moreover, the combined specificity derived from RPA amplification and sgRNA-mediated targeting ensures reliable discrimination between vaccine strains and wild-type viruses. This capability substantially enhances the value of our system in avian disease surveillance and addresses a critical industry need for rapid differential diagnosis of pathogenic MDV strains.

It should be noted that nucleic acid extraction in this study was performed using a commercial kit (TIANamp Virus DNA/RNA Kit), which requires a heating block at 56°C and centrifugation. While suitable for laboratory use, this step may limit true field applicability in settings with restricted access to electricity, heating equipment, or centrifuges. To improve the portability and field readiness of the RPA-CRISPR/Cas14a platform, future work could explore alternative nucleic acid extraction methods better suited to resource-limited environments. For instance, simplified approaches such as lysis buffers combined with filter-based purification, magnetic bead-based extraction, or direct amplification from minimally processed samples (e.g., crude tissue lysates) could be evaluated to achieve fully instrument-free, on-site detection workflows.

In conclusion, a novel visual nucleic acid detection method based on RPA-CRISPR/Cas14a technology was established for MDV. This newly developed assay provides a reliable alternative for MDV detection, requiring minimal equipment for nucleic acid detection after extraction, and shows great potential for application in resource-limited settings.

## ACKNOWLEDGMENTS

We are sincerely grateful to the Institute for Animal Health, Henan Academy of Agricultural Sciences, for their valuable cooperation and support throughout the research process.

This work was supported by the Key Scientific Research Projects of Universities in Henan (no. 22A310017).

Z.-J.Z. and E.-Z.L. contributed to the design of this work. Z.-J.Z., E.-Z.L., and J.-F.L. contributed to the data analysis and interpretation. Z.-J.Z., M.-L.C., Y.L., X.-Q.Y., M.-C.W., J.-H.L., and M.-J.L. contributed to the result validation. Z.-J.Z., E.-Z.L., M.-L.C., Y.L., X.-Q.Y., and J.-F.L. contributed to the drafting and editing of this article. All authors have read and agreed to the published version of the manuscript.

## AUTHOR AFFILIATIONS

[1]Affiliated Central Hospital, Huanghuai University, Zhumadian, People's Republic of China
[2]College of Biological and Food Engineering, Huanghuai University, Zhumadian, People's Republic of China
[3]Topfond Pharmaceutical Co., Ltd., Zhumadian, People's Republic of China
[4]Xiping County Hospital of Traditional Chinese Medicine, Zhumadian, People's Republic of China

## AUTHOR ORCIDs

Zhi-Jian Zhu http://orcid.org/0000-0002-6249-5374
Jin-Feng Li http://orcid.org/0009-0005-8310-5741
En-Zhong Li http://orcid.org/0000-0001-7014-282X

## FUNDING

| Funder | Grant(s) | Author(s) |
|---|---|---|
| Key Scientific Research Projects of Universities in Henan | 22A310017 | Zhi-Jian Zhu |

## AUTHOR CONTRIBUTIONS

Zhi-Jian Zhu, Conceptualization, Data curation, Formal analysis, Funding acquisition, Investigation, Methodology, Project administration, Validation, Writing – original draft, Writing – review and editing | Meng-Li Cui, Data curation, Methodology, Validation, Writing – original draft | Yu Liu, Validation | Xi-Qiao Yao, Validation | Meng-Jie Lu, Validation | Ming-Cheng Wang, Conceptualization, Data curation, Methodology, Writing – review and editing | Jun-He Liu, Conceptualization, Data curation, Methodology, Writing – review and editing | Jin-Feng Li, Conceptualization, Data curation, Methodology, Writing – review and editing | En-Zhong Li, Conceptualization, Data curation, Formal analysis, Methodology, Writing – original draft, Writing – review and editing

## DATA AVAILABILITY

The data generated or analyzed during this study are available from the corresponding author upon reasonable request.

## ETHICS APPROVAL

All experimental protocols were approved prior to the study by the Huanghuai University Experimental Animal Ethics Committee (No. 20240306012).

## ADDITIONAL FILES

The following material is available online.

### Supplemental Material

**Supplemental figures (Spectrum02625-25-s0001.pdf).** Figures S1 to S4.
**Table S1 (Spectrum02625-25-s0002.docx).** Viral loads in clinical samples as determined by qPCR targeting the *pp38* gene.

### Open Peer Review

**PEER REVIEW HISTORY (review-history.pdf).** An accounting of the reviewer comments and feedback.

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
