## [Reviewer comments · Microbiology Spectrum]

Microbiology Spectrum

CRISPR/Cas14a Combined with RPA for Visual Detection of Marek's Disease Virus

Zhi-Jian Zhu, Meng-Li Cui, Yu Liu, Xi-Qiao Yao, Meng-Jie Lu, Ming-Cheng Wang, Jun-He Liu, Jinfeng Li, and Enzhong Li

Corresponding Author(s): Enzhong Li, Huanghuai University

Review Timeline:

Submission Date:	August 23, 2025
Editorial Decision:	November 21, 2025
Revision Received:	December 16, 2025
Editorial Decision:	December 23, 2025
Revision Received:	December 23, 2025
Accepted:	January 9, 2026

Editor: Alexander Bello

Reviewer(s): Disclosure of reviewer identity is with reference to reviewer comments included in decision letter(s). The following individuals involved in review of your submission have agreed to reveal their identity: Grzegorz Woźniakowski (Reviewer #2)

Transaction Report:

DOI: <https://doi.org/10.1128/spectrum.02625-25>

Re: Spectrum02625-25 (CRISPR/Cas14a Combined with RPA for Visual Detection of Marek's Disease Virus)

Dear Prof. Enzhong Li:

Thank you for the privilege of reviewing your work. Below you will find my comments, instructions from the Spectrum editorial office, and the reviewer comments.

Major:

Lines 289-291: How long did it take before samples became positive, on both fluorescence and lateral flow strips? Was RT-qPCR performed on the samples? Are the CT values known for the samples? The researchers need to provide figures for the fluorescence detection and lateral flow strip results. Also, they will need to provide CT values to provide a quantitative measure of the viral genomes present in the sample that can be used to compare to the Cas14a fluorescence and lateral flow detection.

Minor:

Line 146: Is TIANamp Virus DNA/RNA kit appropriate for field use? The kit requires a heating block 56 degrees Celcius and centrifuge. The authors should comment on this in the discussion, regarding the limitations observed with their sample extraction methodology and other alternatives that may be more appropriate for use in the field.

Line 155: Please mention expected PCR fragment size for Meq gene.

Line 220: The meq-F1/R3 product (Lane 3) almost looks the same intensity as the meq-F3/R3 (Lane 9). How did the authors objectively determine that the meq-F1/R3 combination was better than the meq-F3/R3 combination? For Line 221, I would not say the meq-F1/R3 combination resulted in significantly higher amplification unless the authors show this difference by quantitatively measured expression, either by quantitative PCR or DNA concentration.

Figure 3 and 4: the figure legend symbols are too large and masking others, making it difficult to place where some of the samples are in the figure. Please adjust the symbols accordingly so that the samples can be easily identified.

Figure 5: For the negative control and pMD-19T-L-meq strips, it looks like there is a faint line at the test line compared to the sgRNA-free. Please comment on this.

Revision Guidelines

ASM Membership: Corresponding authors may join or renew ASM membership to obtain discounts on publication fees. Need

to upgrade your membership level? Please contact Customer Service at Service@asmusa.org.

Sincerely,
Alexander Bello
Editor
Microbiology Spectrum

Reviewer #1 (Comments for the Author):

Zhu et al. combined RPA isothermal amplification, lambda exonuclease strand targeting and CRISPR/Cas14a detection to develop a molecular diagnostic assay for Marek's disease virus (MDV). The study was well performed and conclusions supported by data. The assay runs at a lower temperature (37 {degree sign}C), has an impressive sensitivity (24.6 copies/ul) and specificity distinguishing pathogenic viral strains from vaccine strains and related viruses, and is available in fluorescent and lateral flow strip formats. It holds great promise for further development for point-of-need application to the diagnosis and surveillance of MDV in resource-limited settings. The study is presented in a short report, simple and clear, which will facilitate good readership.

A major advantage of the described assay is that it achieved a specificity of single nucleotide resolution by utilizing Cas14a. However, the paper should be significantly improved by a better introduction and discussion of the characteristics of this CRISPR enzyme in the context of data from both this study and the literature. Compared to some other Cas proteins such as Cas12, Cas14a has greatly reduced tolerance to mismatches in target sequence recognition. It appeared that the specificity of detection can be defined by a single nucleotide variation such as in this study and in the original paper on this topic (PMID 30337455). It was proposed that there is a seed region in the targeting sequence of Cas14a gRNA where perfect sequence match is required for target recognition. However, other studies such as in PMID 40884799 showed that a single nucleotide mismatch does not always lead to a significant change in target recognition and additional mismatches are needed to facilitate target differentiation. These observations in Cas14a (single strand DNA-targeting) suggest a similarity to Cas13a (single strand RNA-targeting) (PMID 40884799 and PMID 35271569). Collectively, these variations may be reconciled by a position-dependent effect of mismatches on target recognition. The authors should provide a more balanced discussion on this in the context of these papers. This will better avoid a misleading assumption that Cas14 activation always depends on a perfect match between the guide sequence and target sequence.

Other comments:

MATERIALS AND METHODS. To facilitate study reproducibility please provide catalog numbers for reagents used.

Line 114. Regarding "PAM", please provide full form of the term and references for PAM-independence of Cas14a.

Line 117. Please provide references for higher fidelity in target sequence recognition.

Line 118. "This characteristic endows Cas14a with exceptional sensitivity in detecting single-nucleotide polymorphisms (SNPs)." - Should mean exceptional "specificity" as opposed to sensitivity. Need references as well.

Line 468. Degenerate base "N" as "T" or "C". The standard IUPAC ambiguity code for T or C would be Y, while N would represent any of the four bases, A, C, G or T. Please clarify.

Fig. 2. To confirm that the bands represent specific amplicons of interest, results should include at least no-template controls and band sizes while ideally DNA sequencing will confirm the sequences of these bands.

Fig. 4A. The different viral strain templates should be tested at the same molar concentration and a relatively higher concentration to best demonstrate specificity. This should be indicated in methods or figure legends. It also applies to Fig. 6A.

Line 264. "Cas14a-mediated cleavage of the reporter redirects the gold antibody-FAM complexes to the test line,". Please elaborate how the redirection occurs.

Responses to reviewer's comments

Major:

Lines 289-291: How long did it take before samples became positive, on both fluorescence and lateral flow strips? Was RT-qPCR performed on the samples? Are the CT values known for the samples? The researchers need to provide figures for the fluorescence detection and lateral flow strip results. Also, they will need to provide CT values to provide a quantitative measure of the viral genomes present in the sample that can be used to compare to the Cas14a fluorescence and lateral flow detection.

Responses: We sincerely thank the reviewer for these insightful and constructive questions. We have now extensively revised the manuscript to address each point thoroughly, with the corresponding additions and clarifications highlighted below.

1. Detection Time for Fluorescence and Lateral Flow Strips:

We have added detailed information on the detection speed of our RPA-CRISPR/Cas14a platform in the revised “Enhanced Cas14a detection in clinical samples” section (Results, lines 346-350).

For the fluorescence-based assay: Positive signals were typically observed within 20 – 40 minutes, with higher viral loads correlating with faster signal onset.

For the lateral flow assay (LFA): Clear visual test lines appeared within 5 minutes of immersion for all positive samples.

These time points demonstrate the rapidity of our method, making it suitable for on-site diagnostics.

2. Performance of qPCR and Provision of CT Values:

In the revised manuscript, we further performed TaqMan qPCR as an orthogonal, quantitative method to validate our Cas14a-based results and to provide a precise measure of viral load. A new subsection titled “Comparison of RPA-CRISPR/Cas14a with qPCR for clinical sample detection” has been added to the Materials and Methods (lines 224-249) to describe the qPCR protocol in detail.

In the corresponding Results section, we now report: the qPCR standard curve

showed excellent linearity ($R^2 = 0.995$) with the regression equation: $Ct = -3.51 \times \log_{10}(Q) + 36.460$. The detection limit of the qPCR assay was 10.2 copies/ μ L. Critically, the qPCR results were in complete agreement (100% concordance) with both Cas14a-based assays (see updated Table 3), confirming the accuracy of our platform.

3. Quantitative Viral Load Data (Ct values and calculated copies):

To provide the quantitative measure requested by the reviewer, we have:

Normalized viral loads: The absolute pp38 copy numbers were normalized to the cellular content (using the chicken ovo reference gene) and are reported as copies per 10^6 cells. The normalized viral loads for positive clinical samples spanned a wide dynamic range, from 5.2×10^3 to 1.3×10^6 copies per 10^6 cells (see the “Enhanced Cas14a detection...” subsection in Results and Supplementary Table S1).

Ct value availability: The underlying Ct values for all samples (from which these copy numbers were calculated) are provided in Supplementary Table S1, fulfilling the reviewer’s request for the primary quantitative data.

4. Figures for Fluorescence and Lateral Flow Results:

As suggested, we have prepared and now refer to Fig. S2 & S3 (for clinical sample results) and Fig. S4 (for PCR validation) in the revised manuscript. These supplementary figures visually present:

Fig. S2 & S3: Representative results of the fluorescence kinetic curves and lateral flow strips for the panel of clinical samples.

Fig. S4: Gel electrophoresis images confirming the concordance of our method with conventional Meq-PCR.

Minor:

Line 146: Is TIANamp Virus DNA/RNA kit appropriate for field use? The kit requires a heating block 56 degrees Celcius and centrifuge. The authors should comment on this in the discussion, regarding the limitations observed with their sample extraction methodology and other alternatives that may be more appropriate for use in the field.

Responses: We sincerely thank the reviewer for this insightful comment. The reviewer is correct to point out that the nucleic acid extraction method used in this

study (TIANamp Virus DNA/RNA kit) requires laboratory equipment such as a heating block and a centrifuge, which may limit its immediate applicability in resource-limited field settings.

In response to this valuable feedback, we have added a paragraph in the Discussion section (see lines 405 – 415 in the revised manuscript) acknowledging this limitation and discussing potential alternatives for field-compatible nucleic acid extraction. We suggest that future iterations of this diagnostic platform could incorporate simpler, equipment-free extraction methods—such as direct lysis buffers, magnetic bead-based systems, or filter-based purification—to achieve a fully portable, instrument-independent workflow suitable for on-site use.

Line 155: Please mention expected PCR fragment size for Meq gene.

Responses: We sincerely thank the reviewer for this valuable suggestion. As recommended, we have now explicitly stated the expected PCR amplicon size for the Meq gene in the “Generation of DNA standard” section of the revised manuscript (see lines 161 – 162). Specifically, we added: “The expected amplicon sizes were 1,020 bp for Md5 and 1,197 bp for CVI988/Rispens” This addition clarifies the molecular characteristics of the amplified target and enhances the methodological transparency of our study.

Line 220: The meq-F1/R3 product (Lane 3) almost looks the same intensity as the meq-F3/R3 (Lane 9). How did the authors objectively determine that the meq-F1/R3 combination was better than the meq-F3/R3 combination? For Line 221, I would not say the meq-F1/R3 combination resulted in significantly higher amplification unless the authors show this difference by quantitatively measured expression, either by quantitative PCR or DNA concentration.

Responses: We sincerely thank the reviewer for this critical observation. The reviewer correctly points out that the band intensities for primer sets meq-F1/R3 and meq-F3/R3 appear comparable in Fig. 2. Our primary criterion for selecting the optimal primer set was specificity, rather than band intensity alone. Upon closer examination of the agarose gel (Fig. 2, Lane 9), the amplification product from the meq-F3/R3 primer pair exhibited a faint, higher-molecular-weight nonspecific band above the target amplicon. In contrast, the meq-F1/R3 combination (Lane 3) produced

a single, clean band corresponding to the expected size with no detectable nonspecific amplification. Therefore, to ensure assay specificity and reliability in subsequent experiments, we selected the meq-F1/R3 primer set as the optimal combination.

In response to the reviewer's suggestion, we have revised the manuscript to clarify our selection rationale and have toned down the claim regarding "significantly higher amplification." The text now emphasizes the specificity of the meq-F1/R3 primer set as the key deciding factor. (see lines 249 - 252)

Figure 3 and 4: the figure legend symbols are too large and masking others, making it difficult to place where some of the samples are in the figure. Please adjust the symbols accordingly so that the samples can be easily identified.

Responses: We sincerely thank the reviewer for this valuable suggestion regarding the clarity of Figures 3 and 4. In the revised manuscript, we have carefully adjusted the symbol sizes in both figures. (See figure 3 and 4)

Figure 5: For the negative control and pMD-19T-L-meq strips, it looks like there is a faint line at the test line compared to the sgRNA-free. Please comment on this.

Responses: We sincerely thank the reviewer for the careful observation regarding the faint line observed in the negative controls presented in Figure 5. The reviewer is correct that a very faint signal is visible at the test line (T line) position for the negative control and pMD-19T-L-meq samples. In lateral flow assays, such faint background signals can occasionally occur due to non-specific accumulation of the detection components or minimal, non-target-mediated cleavage of the reporter molecule. Crucially, the signal intensity of this background is substantially lower than the strong, definitive T line generated by the specific Cas14a-mediated trans-cleavage in the positive control (pMD-19T-S-meq). This clear visual distinction ensures reliable binary (positive/negative) interpretation of the results, which is further supported by the presence of a robust control line (C line) in all strips, confirming valid test execution.

To address the reviewer's concern with full transparency and to present the clearest possible data, we have replaced the original panel in Figure 5 with a representative replicate from our experimental series that shows no visible background at the T line for the negative controls. All scientific conclusions remain unchanged. The updated figure is provided in the revised manuscript.

Furthermore, to enhance the assay's robustness, we have added a note in the Methods section (under “Cas14a detection with lateral Flow Assay”) recommending the use of freshly prepared reagents and adherence to the specified incubation times to minimize potential non-specific interactions. (see lines 207-209)

We appreciate the reviewer's attention to detail, which has helped us improve the clarity and rigor of our data presentation.

Reviewer #1 (Comments for the Author):

Zhu et al. combined RPA isothermal amplification, lambda exonuclease strand targeting and CRISPR/Cas14a detection to develop a molecular diagnostic assay for Marek's disease virus (MDV). The study was well performed and conclusions supported by data. The assay runs at a lower temperature (37 {degree sign}C), has an impressive sensitivity (24.6 copies/ul) and specificity distinguishing pathogenic viral strains from vaccine strains and related viruses, and is available in fluorescent and lateral flow strip formats. It holds great promise for further development for point-of-need application to the diagnosis and surveillance of MDV in resource-limited settings. The study is presented in a short report, simple and clear, which will facilitate good readership.

A major advantage of the described assay is that it achieved a specificity of single nucleotide resolution by utilizing Cas14a. However, the paper should be significantly improved by a better introduction and discussion of the characteristics of this CRISPR enzyme in the context of data from both this study and the literature. Compared to some other Cas proteins such as Cas12, Cas14a has greatly reduced tolerance to

mismatches in target sequence recognition. It appeared that the specificity of detection can be defined by a single nucleotide variation such as in this study and in the original paper on this topic (PMID 30337455). It was proposed that there is a seed region in the targeting sequence of Cas14a gRNA where perfect sequence match is required for target recognition. However, other studies such as in PMID 40884799 showed that a single nucleotide mismatch does not always lead to a significant change in target recognition and additional mismatches are needed to facilitate target differentiation. These observations in Cas14a (single strand DNA-targeting) suggest a similarity to Cas13a (single strand RNA-targeting) (PMID 40884799 and PMID 35271569). Collectively, these variations may be reconciled by a position-dependent effect of mismatches on target recognition. The authors should provide a more balanced discussion on this in the context of these papers. This will better avoid a misleading assumption that Cas14 activation always depends on a perfect match between the guide sequence and target sequence.

Responses: We sincerely thank the reviewer for the positive and constructive feedback on our manuscript, and particularly for highlighting the crucial need to contextualize the specificity characteristics of Cas14a more thoroughly within the existing literature. We agree that a more nuanced discussion will prevent overgeneralization and provide readers with a balanced understanding.

In direct response to this valuable suggestion, we have significantly expanded both the Introduction and Discussion sections of the revised manuscript.

In the Introduction, following the description of Cas14a's basic mechanism, we have added a paragraph that introduces its renowned sensitivity to mismatches, mentions the “seed region” hypothesis, and importantly, cites recent studies (now references(21, 22) in our list) that indicate the position-dependent and sometimes non-absolute nature of this mismatch sensitivity, drawing a parallel to Cas13a. This sets a more comprehensive theoretical stage for our work. (see lines 118-128)

In the Discussion, we have added a dedicated paragraph that explicitly addresses the reviewer's point. We reaffirm that our observed single-base discrimination is consistent with the high-fidelity paradigm of Cas14a. We then cite the same key studies (21, 22) to acknowledge the documented complexities and position-dependent effects of mismatches on Cas14a activity. We conclude by explaining how our experimental design — combining carefully designed sgRNA with RPA pre-amplification — provided a robust framework that successfully achieved the required specificity in our system, while also suggesting systematic mismatch tolerance studies as a direction for future assay optimization. (see lines 373-385)

We believe these additions provide the necessary depth and balance, accurately placing our findings within the current understanding of Cas14a biology while clarifying the basis for our assay's performance. We are grateful for this insightful comment, which has substantially strengthened the scholarly rigor of our manuscript.

Other comments:

MATERIALS AND METHODS. To facilitate study reproducibility please provide catalog numbers for reagents used.

Responses: We sincerely thank the reviewer for this constructive suggestion. To enhance the reproducibility of our study, we have now supplemented the MATERIALS AND METHODS section with the catalog numbers and manufacturers for key reagents used in the experiments.

Line 114. Regarding "PAM", please provide full form of the term and references for PAM-independence of Cas14a.

Responses: We thank the reviewer for highlighting the need to clarify this important technical term. In response, we have revised the relevant sentence in the Introduction to include the full form of “PAM” and to cite key references that establish the PAM-independent nature of Cas14a. (see line 117)

Line 117. Please provide references for higher fidelity in target sequence recognition.

Responses: We thank the reviewer for this valuable suggestion. In the revised manuscript, we have supplemented the statement regarding Cas14a's high sequence fidelity with appropriate references. (see line 121, References 19, References 20)

Line 118. "This characteristic endows Cas14a with exceptional sensitivity in detecting single-nucleotide polymorphisms (SNPs)." - Should mean exceptional "specificity" as opposed to sensitivity. Need references as well.

Responses: We thank the reviewer for this valuable suggestion. In response, we have revised the relevant sentence (See lines 121) to include appropriate references supporting Cas14a's high sequence fidelity. The updated text now reads:

A key feature of Cas14a is its high sensitivity to single-nucleotide mismatches within the target sequence, which is often linked to a proposed "seed region" in the sgRNA where full complementarity is critical for activation(19, 20). This property makes Cas14a particularly well-suited for distinguishing single-nucleotide polymorphisms (SNPs).

Here, reference [19] is the foundational study by Harrington and Burstein (Science 2018) that first characterized Cas14a's mismatch sensitivity, and reference [20] is a recent study confirming its high-fidelity recognition mechanism. We have also rephrased the sentence as suggested to enhance clarity.

Line 468. Degenerate base "N" as "T" or "C". The standard IUPAC ambiguity code for T or C would be Y, while N would represent any of the four bases, A, C, G or T.

Please clarify.

Responses: We sincerely thank the reviewer for this highly meticulous and accurate observation regarding the IUPAC nucleotide ambiguity codes. In our primer design, we specifically requested the synthesis company to incorporate a degenerate base representing either T or C at the designated position. Our initial description was incorrect. To rectify this inaccuracy and adhere to standard nomenclature, we have revised both Table 1 and its corresponding footnote in the manuscript. The updated note now reads:

Note: "p", 5'-Phosphorylation of primers; "R", Degenerate base (A or G); "Y", Degenerate base (T or C).

We appreciate the reviewer's exceptional attention to detail, which has been instrumental in ensuring the technical precision and clarity of our methodological description.

Fig. 2. To confirm that the bands represent specific amplicons of interest, results should include at least no-template controls and band sizes while ideally DNA sequencing will confirm the sequences of these bands.

Responses: We sincerely thank the reviewer for this important suggestion regarding the validation of the RPA amplification products shown in Figure 2. The reviewer rightly points out that the inclusion of additional controls and molecular weight information would strengthen the evidence for specific amplification.

In response, we would like to clarify that after selecting the meq-F1/R3 primer combination as the optimal RPA primer system, we performed Sanger sequencing of the purified RPA amplification product to confirm the specificity of the selected primer set (meq-F1/R3). The obtained sequence perfectly matched the expected target region within the meq gene of the MDV-1 Md5 strain, confirming the absence of nonspecific amplification. A statement to this effect has been added to the “Optimal RPA primer set selection” section of the Results chapter (See lines 260 – 263). Additionally, information on the size of the amplification band produced by the meq-F1/R3 primer set has been added to the Results section (See lines 259). Furthermore, we have included a no-template control (NTC) for all nine primer combinations tested, and agarose gel electrophoresis confirmed that no amplification bands were observed in any of these negative control reactions (data not shown).

We believe these revisions provide the necessary validation for the experiment and significantly enhance the reliability of the data presented in Figure 2. We are grateful to the reviewer for this constructive feedback, which has made our findings clearer and more rigorous.

Fig. 4A. The different viral strain templates should be tested at the same molar concentration and a relatively higher concentration to best demonstrate specificity.

This should be indicated in methods or figure legends. It also applies to Fig. 6A.

Responses: We sincerely thank the reviewer for this insightful suggestion, which is crucial for rigorously demonstrating the specificity of our assay. We fully agree that testing different viral templates at an equimolar and relatively high concentration provides the most convincing evidence for specificity. Our experiments were indeed conducted precisely under these conditions.

In direct response to this comment, we have revised the manuscript as follows:

In the RESULTS section, we have added a clear statement specifying that for specificity evaluations, nucleic acid extracts from all tested viruses (MDV-1 Md5, CVI988/Rispens, HVT, NDV, and IBDV) were adjusted to an equimolar concentration of 10^6 copies/ μL prior to use in the RPA-CRISPR/Cas14a assays (see lines 279–281 in the revised manuscript).

In the Figure Legends for both Fig. 4A and Fig. 6A, we have explicitly noted that “All viral nucleic acid templates were tested at an equimolar concentration of 10^6 copies/ μL .”

We are grateful for the reviewer’s guidance in enhancing the rigor of our data presentation.

Line 264. "Cas14a-mediated cleavage of the reporter redirects the gold antibody-FAM complexes to the test line,". Please elaborate how the redirection occurs.

Responses: We thank the reviewer for this request for clarification on the lateral flow detection mechanism. In response, we have revised the relevant sentence in the “Validation of the enhanced Cas14a lateral flow detection” section to provide a more detailed explanation of the redirection process. (See lines 306 – 313)

This revision clarifies that the cleavage event physically separates the reporter molecule, allowing the FAM-labeled fragment to bind to the gold-antibody conjugate and subsequently be captured at the test line. We appreciate the reviewer's attention to detail, which has helped improve the clarity of our methodology description.

Re: Spectrum02625-25R1 (CRISPR/Cas14a Combined with RPA for Visual Detection of Marek's Disease Virus)

Dear Prof. Enzhong Li:

Thank you for the privilege of reviewing your work. Below you will find my comments, instructions from the Spectrum editorial office, and the reviewer comments.

Thank you for addressing my comments. However, there are a few additional minor comments that need to be addressed:

Lines 42 and 54: ...instrument-free... - this is misleading and too strongly worded as the methodology presented in this paper isn't entirely instrument-free. Nucleic acid extraction from the sample still required a centrifuge based on the methodology used in this paper and instrument-free nucleic acid extraction methods were not evaluated. It's true that the nucleic acid detection is instrument-free, but the authors need to indicate that the assay was performed on previously extracted nucleic acid or remove the instrument-free description.

Lines 54, 240, 265, 301, 393: Clinical validation - CLSI recommends {greater than or equal to} 50 positive samples and {greater than or equal to} 100 negative samples for accuracy studies, and 20 replicates at the claimed limit of detection for a proper validation. A reference range with a minimum of 20 samples that are representative of the population would also need to be established for a clinical validation. The authors should not use the word (clinical) validation in their manuscript if a proper validation was not performed. This study is more like an evaluation. Only 17 positive and 7 negative samples were included in this study, which doesn't meet the same rigor required by a proper clinical validation. Please change wording accordingly, such as to evaluate or evaluation.

Line 295: Should specify that this is an "analytical sensitivity" since this is determining limit of detection.

Revision Guidelines

Sincerely,
Alexander Bello
Editor
Microbiology Spectrum

Reviewer #1 (Comments for the Author):

The revised manuscript has well addressed my questions, comments and suggestions. I have no further additions.

Responses to reviewer's comments

Lines 42 and 54: ...instrument-free... - this is misleading and too strongly worded as the methodology presented in this paper isn't entirely instrument-free. Nucleic acid extraction from the sample still required a centrifuge based on the methodology used in this paper and instrument-free nucleic acid extraction methods were not evaluated. It's true that the nucleic acid detection is instrument-free, but the authors need to indicate that the assay was performed on previously extracted nucleic acid or remove the instrument-free description.

Responses: We sincerely thank the reviewer for the insightful comment regarding the use of the term “instrument-free” in our manuscript. We agree that the term could be misleading, as nucleic acid extraction was performed using a commercial kit requiring a centrifuge and heating block.

We have now revised the manuscript to clarify that the detection step (RPA-CRISPR/Cas14a amplification and readout) is instrument-free and suitable for visual or lateral-flow-based interpretation, while the nucleic acid extraction step in this study was performed with standard laboratory equipment. The term “instrument-free” has been replaced with more precise phrasing such as “without complex detection instruments” or “visual readout without instrumentation” in the relevant sections (Lines 42 - 43, 55 - 56, and 421).

Thank you for helping us improve the clarity and accuracy of our manuscript.

Lines 54, 240, 265, 301, 393: Clinical validation - CLSI recommends {greater than or equal to} 50 positive samples and {greater than or equal to} 100 negative samples for accuracy studies, and 20 replicates at the claimed limit of detection for a proper validation. A reference range with a minimum of 20 samples that are representative of the population would also need to be established for a clinical validation. The authors should not use the word (clinical) validation in their manuscript if a proper validation

was not performed. This study is more like an evaluation. Only 17 positive and 7 negative samples were included in this study, which doesn't meet the same rigor required by a proper clinical validation. Please change wording accordingly, such as to evaluate or evaluation.

Responses: We sincerely thank the reviewer for the valuable and rigorous comment regarding the use of the term “clinical validation” in our manuscript. We fully acknowledge that our study, with 17 positive and 7 negative clinical samples, does not meet the formal sample size requirements for a comprehensive clinical validation as outlined in guidelines such as CLSI.

Accordingly, we have revised the manuscript to replace terms such as “validation” with more appropriate descriptors like “evaluation,” “assessment,” and “performance testing” in all relevant sections (Lines 55, 243, 267, 303, 395, and elsewhere). We have also adjusted the corresponding section headings and narrative to reflect that this work represents a preliminary clinical evaluation rather than a full clinical validation.

We hope these revisions provide a more accurate representation of the scope and intent of our study, which aimed to demonstrate the feasibility and diagnostic potential of the RPA-CRISPR/Cas14a platform in a limited clinical sample set. We appreciate your guidance in improving the precision of our terminology.

Line 295: Should specify that this is an "analytical sensitivity" since this is determining limit of detection.

Responses: Thank you for the valuable suggestion regarding the clarification of sensitivity terminology. We have revised the sentence to explicitly state that the evaluation pertains to analytical sensitivity (Lines 297).

This revision ensures clarity and aligns with common terminology used in assay performance characterization. We appreciate your attention to this detail.

Re: Spectrum02625-25R2 (CRISPR/Cas14a Combined with RPA for Visual Detection of Marek's Disease Virus)

Dear Prof. Enzhong Li:

Your manuscript has been accepted, and I am forwarding it to the ASM production staff for publication. Your paper will first be checked to make sure all elements meet the technical requirements. ASM staff will contact you if anything needs to be revised before copyediting and production can begin. Otherwise, you will be notified when your proofs are ready to be viewed.

Sincerely,
Alexander Bello
Editor
Microbiology Spectrum